# Impaired Mitochondrial Function in T-Lymphocytes as a Result of Exposure to HIV and ART

**DOI:** 10.3390/cells12071072

**Published:** 2023-04-02

**Authors:** Alexander V. Blagov, Vasily N. Sukhorukov, Shuzhen Guo, Dongwei Zhang, Mikhail A. Popov, Alexander N. Orekhov

**Affiliations:** 1Laboratory of Angiopathology, Institute of General Pathology and Pathophysiology, 8 Baltiiskaya Street, Moscow 125315, Russia; 2Diabetes Research Center, School of Traditional Chinese Medicine, Beijing University of Chinese Medicine, Beijing 100029, China; 3Department of Cardiac Surgery, Moscow Regional Research and Clinical Institute (MONIKI), 61/2, Shchepkin Street, Moscow 129110, Russia

**Keywords:** mitochondrial dysfunction, HIV, AIDS, CD4+T-lymphocytes

## Abstract

Mitochondrial dysfunction is a described phenomenon for a number of chronic and infectious diseases. At the same time, the question remains open: is this condition a consequence or a cause of the progression of the disease? In this review, we consider the role of the development of mitochondrial dysfunction in the progression of HIV (human immunodeficiency viruses) infection and the onset of AIDS (acquired immunodeficiency syndrome), as well as the direct impact of HIV on mitochondria. In addition, we will touch upon such an important issue as the effect of ART (Antiretroviral Therapy) drugs on mitochondria, since ART is currently the only effective way to curb the progression of HIV in infected patients, and because the identification of potential side effects can help to more consciously approach the development of new drugs in the treatment of HIV infection.

## 1. Introduction

HIV (human immunodeficiency virus) is a virus from the family of retroviruses that have double-stranded RNA in their composition, which determines the nature of infection and the spread of this virus. HIV infects T-helpers, and not only infects a cell and uses its resources to generate viral progeny, but also, thanks to the possibility of reverse transcription, integrates its own genome into the genome of the host cell. This feature makes the virus invisible and, given that the primary symptoms of HIV are not specific to this virus, an infected person often becomes a hidden reservoir for the spread of HIV. This feature is one of the key reasons why HIV has spread throughout the world and in many countries and has acquired the status of an epidemic disease. For 2021, according to WHO statistics, there were 38.4 million people infected with HIV in the world, of which 1.5 million learned about their diagnosis in 2021 [1]. HIV is dangerous because its presence in the body leads to a sharp decrease in the number of CD4 + T-lymphocytes, resulting in a state of acquired immunodeficiency, AIDS, in which the concentration of T-lymphocytes is 200 cells per microliter or less [2]. AIDS contributes to the development of opportunistic infections in the body that are usually not manifested in a normal immune status. The most common opportunistic infections in AIDS include tuberculosis, candidiasis, herpes and cytomegalovirus infection [3]. In addition, AIDS is associated with the development of malignant tumors such as cervical cancer, Kaposi’s sarcoma and non-Hodgkin’s lymphoma [4]. Without the use of specialized antiviral treatment—antiretroviral therapy (ART)—life expectancy in patients with AIDS is no more than 2 years after diagnosis [5]. As already mentioned, the predominantly latent nature of the infection in the first years after the disease complicates the timely diagnosis of HIV infection, which contributes to the transition of the disease from the chronic phase to AIDS. Currently, there are several tests based on the detection of HIV antibodies, antigens and RNA; however, in the earliest stages of the disease, RNA detection is the most reliable method [6]. Despite the existing shortcomings, the introduction of ART therapy into clinical practice contributed to slowing down the progression of the disease and improving the condition of HIV-infected patients. Undergoing ART prolongs the lives of AIDS patients by more than 10 years [5], while the lack of this treatment leads to death within 2 years, as noted above. The main goal of ART is to block the reproduction of HIV. ART is administered as a simultaneous combination of drugs for which action is based on an inhibitory intervention at different stages of the HIV life cycle. The main classes of ART drugs are nucleotide and non-nucleotide reverse transcriptase inhibitors, protease inhibitors, fusion inhibitors, CCR5 receptor antagonists, T-helper receptors, integrase inhibitors, post-attachment inhibitors and pharmacokinetic enhancers [7]. Although ART is a necessary intervention to prevent the progression of HIV, the presence of many side effects creates the preconditions for the development of new anti-HIV drugs. To reduce side effects in the future, it is important to understand the cellular and molecular mechanisms underlying their occurrence. Drugs, such as antibiotics, are known to be toxic to cell organelles such as mitochondria [8]. Since the mitochondria are of great importance to maintaining cellular energy supply, and, accordingly, to the survival of the cell, any negative impact that leads to the loss or weakening of mitochondrial function, the emergence of dysfunction, can cause cell death and the development of other pathological processes. Indeed, the role of mitochondrial dysfunction has been confirmed in the pathogenesis of a number of chronic diseases [9]. In addition, it is known that the mitochondrion is a target for many viruses, which determines the role of mitochondrial dysfunction in the development of infectious diseases [10]. HIV infection is both an infectious and chronic disease, which indicates a high probability of the involvement of mitochondrial dysfunction in the pathogenesis of this disease. Understanding the role of mitochondrial dysfunction in the progression of HIV infection can be an important step in the fight against this disease. This review has two important goals: first, to develop several pathophysiological models of the impact of HIV on mitochondria that address various aspects of this effect, which will help to better understand the pathogenesis of HIV infection and, second, to investigate the side effects of modern antiretroviral drugs that are associated with the development of mitochondrial dysfunction, which will contribute to a better understanding of the potential side effects that may occur with new antiretroviral drugs. To date, there are a number of studies and reviews on this topic. The closest review might be [11], which also described the impact of HIV and ART on the development of mitochondrial dysfunction, but the authors of that review focused more on the impact of individual HIV proteins on the development of mitochondrial dysfunction. The current review focuses primarily on the complex impact of HIV on various aspects of mitochondrial activity, such as mitochondria-mediated apoptosis, energy metabolism, antiviral immune response, calcium transport and mitochondrial dynamics. Knowing these aspects improves our understanding of what the virus wants to obtain from the cell, by using and modifying the work of its systems. The effect of ART drugs on mitochondria also affects the fate of the cells in which they reside. Thus, in [12], the effect of ART drugs on the development of mitochondrial dysfunction in macrophages was studied. This exposure to ART was found to eventually lead to a preferential polarization of macrophages towards a pro-inflammatory phenotype.

## 2. HIV Pathogenesis

### 2.1. Acute Phase

Despite the fact that HIV can multiply to a limited extent in dendritic cells, HIV has a primary tropism for CD4 + T-lymphocytes expressing the CCR5 receptor [13]. When infected through the mucous membranes (genital organs and anus), the virus undergoes local spread due to the concentration of target cells on the mucous membranes (especially in the intestines), after which the virus, similarly to the cells infected by it, can migrate to the lymph nodes and other organs [14]. When infected through the bloodstream, the virus does not encounter a local concentration of T-helpers; therefore, it immediately migrates to all organs [14]. In a model of monkeys infected with simian immunodeficiency virus, it was shown that viremia is detected 2–3 days after intravenous infection, and the maximum value is recorded 10–14 days after infection, by which time the virus can be detected in all lymphoid organs [15]. At 2–3 weeks after infection, it is found that, in addition to T-helpers, macrophages become another population of infected cells in lymphoid organs [14]. There is a hypothesis that macrophages serve as the most effective carriers for the systemic spread of HIV throughout the body [16]. At the same time, there is an accumulation of HIV virions in follicular dendritic cells. This concentration of virions, according to one hypothesis, is also one of the main reservoirs of HIV in the body [17]. It is believed that it is during the acute phase that a sharp decrease in immune competence occurs, caused by increased HIV replication and the intensive destruction of CD4 + T-lymphocytes by this replication, and that, during the chronic phase, immune activity remains close to a critical level, gradually fading [18]. Such a sharp reduction in the population of CD4 + T-lymphocytes leads to a change in the normal immune function, not only in relation to HIV but also in relation to other pathogens. The innate immune response develops 3–5 days after infection and is accompanied by an increased release of pro-inflammatory cytokines and interferon alpha. The initial producers of cytokines are dendritic cells, but they are then joined by macrophages, natural killers and T-lymphocytes [16]. However, in the case of the increased production of cytokines, a cytokine storm can develop, which can have a detrimental effect on T-helpers. The pool of neutralizing antibodies occurs only 3 months after the onset of infection, which is too late of a protective reaction to change the course of the disease [14]. CD8+ T-lymphocytes act as the main defense against HIV infection. It has been shown that T-killers are able to reduce the viral load in plasma in the early stages of the disease, thus controlling and slowing down the dissemination of HIV [19]. Thus, in a model of rhesus monkeys infected with simian immunodeficiency virus, it was shown that the depletion of CD8+ T-lymphocytes leads to the progression of the disease [20]. Despite this, a strong cytotoxic immune response is not a sufficient measure for the elimination of HIV.

### 2.2. Chronic Phase and AIDS

The chronic phase is the longest period of HIV infection, which can last several years. Moreover, this phase is also called asymptomatic due to the predominant absence of clinical symptoms. During the chronic phase, HIV continues to amplify, and the number of combat-ready T-helpers slowly decreases to critical values. The number of viral particles in this situation can vary greatly in different patients; however, regardless of the presence in the plasma, virions are always present in peripheral blood cells [21]. At the same time, it is noted that the polarization of the immune response shifts from the Th1 subtype to Th2, which may be due to the predominant depletion of Th1 lymphocytes [22]. This increases the concentration of anti-inflammatory cytokines, such as IL10, which inhibit the proliferation of lymphocytes, further accelerating the onset of an immunodeficiency state. The cycles of death and proliferation of T-helper cells ultimately end in the development of a state of immunodeficiency due to a decrease in the ability of naive memory cells to regenerate. In addition, a decrease in the reserve of T-lymphocytes may also be associated with the direct cytotoxic effect of HIV, apoptosis induced by the virus, immunological aging, and the destruction of lymphoid organs due to fibrosis caused by the action of various cytokines [19]. When the concentration of T-helpers reaches less than 200 cells per microliter, the disease passes to the final stage, called AIDS. At the same time, the introduction of ART at this stage can prevent the development of AIDS by suppressing HIV replication, which leads to a decrease in viral load and an increase in the number of CD4 + T-lymphocytes [16]. The key problem of the state of AIDS is the increased sensitivity of the body to various diseases. Those diseases, in turn, can positively affect the progression of HIV infection. Thus, for a number of infectious diseases, such as herpes virus type 2, hepatitis C and tuberculosis, it was demonstrated that their coinfection with HIV leads to an increase in HIV replication [19]. In AIDS, in the absence of proper treatment, and in contrast to the chronic phase of infection, the number of T-helpers decreases linearly, and not in waves, which is associated with the exhaustion of the proliferative potential of memory T-cells [21].

## 3. The Role of Mitochondria in Antiviral Immunity

### 3.1. The Role of Mitochondria in Cells

Mitochondria are double-membrane organelles that contain an autonomous genome represented by a circular DNA molecule, which is one of the prerequisites for the theory of the bacterial origin of mitochondria. The main function of mitochondria in the cell is associated with the generation of energy for cellular needs, which is accumulated in the form of ATP molecules during the process of oxidative phosphorylation. In addition to their direct energy function, mitochondria serve as a hub for various metabolic reactions, thus taking part in the metabolism of carbohydrates, amino acids and lipids, as well as various cofactors. In addition, mitochondria are centers for the transmission of various cellular signals. They are involved in the buffering of calcium ions, the processes of cellular aging and apoptosis [23]. One of the most recently discovered roles associated with the functions of mitochondria is their participation in innate antiviral immunity in vertebrates. This function opens up mitochondria from a new perspective, presenting them as one of the regulators of the body’s immune defense [23]. In this regard, it is interesting to consider what effect the viruses themselves have on mitochondria in order to reduce the strength of the immune response, as well as the possibility of mitochondrial antiviral proteins as targets for antiviral therapy.

### 3.2. Mechanism of Antiviral Immune Response Involving Mitochondria

Triggering the innate immune response against RNA viruses involves the activation of several signaling steps, which ultimately initiate the rapid production of pro-inflammatory cytokines. One of the main types of cytokines are type I interferons, such as IFN-α and -β [24]. Pro-inflammatory cytokines activate the proliferation and migration of leukocytes to the focus of inflammation containing viral particles. Presentation of cleaved antigens by dendritic and other antigen-presenting cells to T-lymphocytes leads to the triggering of an adaptive antiviral immune response. Primary signal transduction is mediated by two distinct pathways: one is triggered by viral RNA binding to endosomal Toll-like receptor 3 (TLR-3), and directed to RNA viruses entering the cell during endocytosis; the other is activated when viral RNA binds to retinoic acid-inducible gene I (RIG-I)-like receptors (RLRs), which are able to recognize viruses with double-stranded RNA located in the cytoplasm [23]. The antiviral response involving mitochondria occurs as a result of the activation of the RLR signaling pathway. At the same time, the central regulator in this process is a protein, mitochondrial antiviral signaling (MAVS), also known under the designations VISA and Cardif [25]. The involvement of the MAVS protein in antiviral signaling has been proven in experiments on mice with a knockout of the MAVS gene, which stops the production of type I interferons and other pro- inflammatory cytokines [26]. As a result of the interaction of RIG-I with double-stranded viral RNA, the complex of combined RIG-I and RNA molecules is translocated to the outer mitochondrial membrane, where binding to MAVS occurs via the interaction of CARD domains. MAVS then oligomerizes and activates signaling through downstream signaling molecules. such as the TRAF family proteins, which ultimately leads to the activation of transcription factors NF-κB and IRF-3 upon completion of the signaling chain, which then initiates the increased expression of pro-inflammatory cytokines [23]. It is also known that there is not only a mitochondrial MAVS system but also a peroxisomal MAVS system that complements it, which was proved in the study [27]. Thus, peroxisomal MAVS elicits a rapid, but transient, antiviral response that inhibits the spread of the virus until mitochondrial-localized MAVS induced a greater sustained antiviral response. Peroxisomal MAVS is not able to directly induce the production of type I interferon, but does cause the secretion of other antiviral proteins, such as viperin and ISG15, as a result of the early induction of interferon-stimulated genes (ISGs) [28]. Thus, with the coordinated work of both systems, the effectiveness of the antiviral response increases. The general scheme of the mitochondrial and peroxisomal MAVS systems is shown in Figure 1.

### 3.3. Mitochondrial and Viral Regulators of MAVS

Mitochondria themselves contain a number of proteins that inhibit the action of the MAVS-associated antiviral immune response. Thus, the NLRX1 protein is highly expressed in mitochondria. One of its functional activities is the inhibition of the CARD–CARD interaction between RLR and MAVS, which inhibits antiviral signaling even at the initial stages [29]. Another mitochondrial inhibitor of MAVS is the gC1qR protein, which is a receptor for the C1q complement protein [30]. Another mitochondrial inhibitor of MAVS is the Mfn2 protein, which has a main function that is related to mitochondrial fusion [31]. The described inhibition of MAVS is physiological and is most likely due to the suppression of the increased inflammatory response caused by the production of pro-inflammatory cytokines. At the same time, another mitochondrial fusion protein, Mfn1, performs the opposite function and activates MAVS [23]. In contrast to the physiological inhibition of the MAVS pathway involving mitochondrial proteins, some viruses are also able to inhibit the action of MAVS; however, it is clear that the result of such inhibition is different and is intended to directly reduce the antiviral immune response and increase the production of virions in the cell. Thus, for hepatitis C viruses and SARS-CoV, it was shown that their proteins are able to inhibit the action of MAVS, as a result of which, in addition to triggering the immune response, apoptosis is also inhibited (since MAVS is also involved in mitochondria-mediated apoptosis), which favorably affects the multiplication virions and the further spread of the virus through body tissues [32]. The MAVS regulation is shown in Figure 2. The role HIV has on MAVS and mitochondria in general will be described in the next section.

## 4. Impact of HIV on the Development of Mitochondrial Dysfunction

A number of studies have shown that mitochondrial activity changes during HIV infection. It was found that the transmembrane mitochondrial potential decreases in patients with HIV infection who did not receive ART compared to people not infected with HIV, and the size of the mitochondrial potential was negatively dependent on the number of lymphocytes undergoing apoptosis [33]. The results of this study indicate that the impairment of mitochondrial function is likely due to the inhibitory effect of HIV proteins.

### 4.1. Triggering Mitochondria-Mediated Apoptosis

Important HIV proteins required for viral replication and integration are structural proteins, including Gag, Pol and Env, in addition to a number of regulatory proteins, including Nef, Vpr, Vif, Vpu, Tat and Rev. It is noted that some of these proteins activate mitochondria-mediated apoptosis [34]. At the same time, apoptosis caused by viral proteins spreads to neighboring cells as a result of the increased expression of these HIV proteins. This hypothesis explains how HIV is able to cause T-helper depletion at a rate that exceeds the rate of virion propagation [35]. Let us consider this mechanism using the Env glycoprotein as an example, which is cleaved into 2 proteins during maturation: gp41 and gp120. The key role in the initiation of apoptosis is played by gp120, which is associated with interactions with co-receptors of CD4+ T-lymphocytes, CXCR4 and CCR5. This interaction ensures the penetration of HIV into the target cell. At the same time, it has been shown that gp120 is able to induce mitochondria-mediated apoptosis by depolarizing the mitochondrial membrane, which primarily leads to a disruption in the process of oxidative phosphorylation and, as a result, also leads to a decrease in ATP production and the energy depletion of T-lymphocytes, finally leading to cell death. Depolarization of the mitochondrial membrane leads to the subsequent rupture of the outer membrane and the release of mitochondrial proteins into the cytoplasm, among which cytochrome C is present, which activates the apoptosis pathway through the activation of caspases 3 and 9 [11]. Due to its ability to bind to cellular receptors, localize to the cytoplasmic membrane of an infected cell and transfer to an adjacent uninfected cell, Env can activate mitochondrial dysfunction and lymphocyte apoptosis without direct infection with virions, which is one of the explanations for the rapid decline in CD4 + T-lymphocytes at an early stage of infections [36]. The general scheme for triggering mitochondria-mediated apoptosis induced by HIV proteins is shown in Figure 3.

### 4.2. Inhibition of the Immune Response against HIV

As reported in the previous section, the mitochondrion, or, rather, its signaling protein, MAVS, is the initiator of a cascade that triggers an innate antiviral immune response directed against double-stranded RNA viruses. Since HIV is a double-stranded RNA virus, it is logical to assume that there is a high probability of close interaction between HIV proteins and the mitochondrial MAVS protein, which can change activity in both directions. Indeed, a study [37] showed that MAVS inhibited HIV proliferation, by triggering interferon response genes, and activated dendritic cell maturation. At the same time, the possibility of avoiding the HIV antiviral immune response caused by dendritic cells has been shown. In another study [38], it was found that HIV binds to the DDX3 RNA helicase, which acts as an HIV RNA receptor. As a result, through the activation of the antiviral signaling pathway, in which MAVS plays a central role, the maturation of dendritic cells and the production of interferon and other cytokines occurred. It was observed that the binding of HIV to the C-type lectin 1 receptor DC-SIGN led to the activation of the mitotic kinase PLK1, which suppressed signaling after MAVS, weakening antiviral protection. Thus, changing the preferred receptor for HIV RNA binding is of great importance to the activation of the innate immune response against HIV. The scheme of inhibition of the immune response against HIV is shown in Figure 4.

### 4.3. Impact on Mitochondrial Dynamics

To maintain the correct energy state in the cell, mitochondria must be in balance with each other and with the processes taking place in the cell by regulating their quality control, quantity and location. It has been shown that the HIV Nef protein is able to inhibit the mitophagy process, which is directly responsible for the quality of mitochondria [39]. This disruption leads to the accumulation of a large proportion of damaged and dysfunctional mitochondria in the cell. This affects the occurrence of energy starvation in the cell, as well as an increase in the production of reactive oxygen species, further damaging mitochondria, including mitochondrial DNA, which, in turn, leads to the spread of the number of dysfunctional mitochondria in the cell and the initiation of apoptosis. The HIV Vpr protein has been shown to have an inhibitory effect on the regulatory protein of mitochondrial dynamics Mfn2, which plays an important role in mitochondrial fusion, transport and contact with the endoplasmic reticulum (ER) [40]. In [41], a study on pregnant women infected with HIV, a decrease in the expression of Mfn2 and an increase in the expression of the apoptosis regulatory protein caspase 3/β were demonstrated, which was explained by the influence of the Vpr protein. Vpr can directly bind to the ER membrane and to the outer mitochondrial membrane. The inhibition of mitochondrial fusion can hamper the repair of damaged mitochondria (because the fusion of unhealthy and healthy mitochondria results in one healthy elongated mitochondria through redistribution of components), as well as an increase in the number of short mitochondria, which quickly exhaust their potential and become dysfunctional. Furthermore, the disruption of the interaction between mitochondria and ER leads to impaired calcium transport in the cell, which can cause ER stress in the form of a response to unfolded proteins (UPR) and, if the response is strong, cause the death of lymphocytes. It was also shown that HIV Tat protein causes an increase in the activation of the mitochondrial division regulator DRP1 on astrocyte cell culture, which led to neurodegeneration [42]. The impact on the processes of mitochondrial dynamics leads to a general disruption of the mitochondrial chain and a more rapid development of pathological events, in contrast to the impact on single mitochondria. The general scheme of the impact of HIV on mitochondrial dynamics is presented in Figure 5.

## 5. Impact of ART on the Development of Mitochondrial Dysfunction

Since ART is currently the only effective way to treat HIV and AIDS, it is important to identify side effects and mechanisms for the development of emerging disorders. Understanding these processes is essential for the development of modified, improved HIV drugs with fewer side effects. Unfortunately, in addition to the direct harmful effects of HIV on the structure and function of mitochondria, as demonstrated in the previous section, some of the currently used ART drugs can also cause mitochondrial dysfunction, which is an additional risk factor for the development of complications from this disease.

### 5.1. Effect of Nucleoside Reverse Transcriptase Inhibitors on Mitochondria

The first drugs approved for the treatment of HIV were nucleoside reverse transcriptase inhibitors (NRTIs). The mechanism of action of NRTIs is based on their insertion into HIV DNA and the initiation of a chain break, which leads to a disruption of the process of reverse transcription of HIV RNA [43]. NRTI drugs have been found to have an inhibitory effect on DNA polymerase gamma. DNA polymerase gamma is involved in the replication and repair of mitochondrial DNA, which is the first evidence of the occurrence of mitochondrial dysfunction under the action of ART drugs [44]. NRTIs are able to integrate into mitochondrial DNA through their interaction with DNA polymerase gamma, which ultimately leads to the appearance of mutations and the inhibition of mitochondrial DNA synthesis. Another mechanism of action of NRTIs on mitochondria is the disruption of ATP/ADP translocation, which disrupts energy metabolism in cells [45]. Additionally, it has been shown that NRTIs reduce the expression of cytochrome c oxidase, which also leads to the disruption of oxidative phosphorylation and cellular respiration in general [46]. The resulting disturbances in oxidative phosphorylation lead to an increase in the release of reactive oxygen species, which contributes to the development of oxidative stress and further damage to mitochondrial proteins, DNA and lipids [47].

### 5.2. Effect of Non-Nucleoside Reverse Transcriptase Inhibitors on Mitochondria

Other classes of ART drugs, including non-nucleoside reverse transcriptase inhibitors (NNRTIs) and protease inhibitors (PIs), have also been shown to inhibit mitochondrial function. NNRTIs inhibit HIV reverse transcription by binding to a hydrophobic pocket immediately adjacent to the active site of the enzyme [48]. NNRTIs, such as Efavirenz, do not directly affect mitochondrial DNA, but do most likely affect proteins of the outer mitochondrial membrane, which leads to a decrease in the electrochemical potential of the mitochondrial membrane, followed by mitochondrial dysfunction, cytochrome release and the activation of apoptosis [49]. It has been shown that the NNRTI-induced impairment of mitochondrial function can be restored by the antioxidant drug Trolox, which indicates a significant role of oxidative stress in the spread of mitochondrial dysfunction in the cell [49].

### 5.3. Effect of Protease Inhibitors on Mitochondria

The mechanism of action of IP is associated with the inhibition of the maturation of HIV proteins by blocking the cleavage of the polypeptide precursors of these proteins as a result of the inhibitory effect on HIV protease [48]. The action of PIs on mitochondria is based, as in the case of NNRTIs, on a decrease in the potential of the mitochondrial membrane, the development of oxidative stress and, accordingly, a violation of oxidative phosphorylation and the induction of apoptosis [50]. These disorders at the macro level can lead to the development of cardiovascular diseases in patients with HIV [51]. Additional effects of PI are associated with an increase in the expression of NADPH oxidase, an enzyme that produces superoxide radical, which further enhances oxidative stress [52], as well as metabolic disorders associated with glucose transport and lipid metabolism [53]. The general scheme of the effect of ART drugs on the development of mitochondrial dysfunction in HIV is presented in Table 1.

## 6. Future Research

There are still many unsolved questions regarding the impact of ART and HIV on the functions of mitochondria. While, for HIV virions, it is clear which cells are affected and which acquire mitochondrial dysfunction due to the specialized tropism of the virus, for ART drugs, this is not so obvious, since the possibility of induction of mitochondrial dysfunction on various cell lines has been shown [49,53]. Knowing in which types of cells, and, accordingly, tissues and organs, drugs cause mitochondrial dysfunction is important for revealing the emerging side effects and complications from therapy at the macro level. This knowledge will allow for the following: first, to assess the real danger to the body that may arise as a result of drug-induced mitochondrial dysfunction; second, to develop new ART drugs that will have a safer profile compared to conventional therapy; third, to assess the prospect of using drugs that restore mitochondrial dysfunction, as shown with the introduction of the antioxidant drug, Trolox [49]. As for HIV itself, it is important to consider the appropriateness of using drugs that restore mitochondrial function, as well as to what extent these therapeutic agents will help improve the condition of patients with HIV, reduce the production of HIV virions and prevent a further decrease in the level of CD4+ T-lymphocytes. Potential strategies in this area include the use of compounds that increase the activity of the cellular antioxidant system, enhance mitochondrial biogenesis and modulate mitochondrial dynamics in the direction of enhancing fusion and weakening fission [54]. In addition to this, the topic related to the use of protective systems by mitochondria to maintain effective work in conditions of HIV infection requires more study. A better understanding of cellular “self-defense” under conditions of viral invasion may provide new therapeutic protein targets, the activation of which can potentially alleviate the condition of HIV patients. Finally, it is also important to study and compare the development of mitochondrial dysfunction in the course of the disease, from the first stage of HIV infection to AIDS. This will help determine the “working capacity” of mitochondria, depending on the level of cell damage by the virus. The possibility of developing the same negative effect both from the action of the virus and from the action of the drug against which it is directed can be a serious condition contributing to the further spread of HIV and the progression of the disease. Thus, this topic requires further close study.

## 7. Conclusions

Mitochondrial dysfunction is one of the hallmarks of HIV infection. Various HIV proteins are able to interact with mitochondria in different ways, which leads to the direct induction of the internal pathway of apoptosis, the disruption of oxidative phosphorylation, the development of oxidative stress, and the disruption of mitochondrial dynamics, including mitophagy and mitochondrial fusion, calcium buffering, and antiviral immune response. The development of mitochondrial dysfunction is one reason to explain the rapid depletion of CD4+ T-lymphocytes in the acute phase of infection. Different types of ART drugs can also cause mitochondrial dysfunction through different mechanisms. To further understand the consequences of mitochondrial dysfunction caused by ART drugs, further studies are required to assess the relationship of mitochondrial dysfunction with the resulting side effects in HIV-infected patients.

## Figures and Tables

**Figure 1 cells-12-01072-f001:**
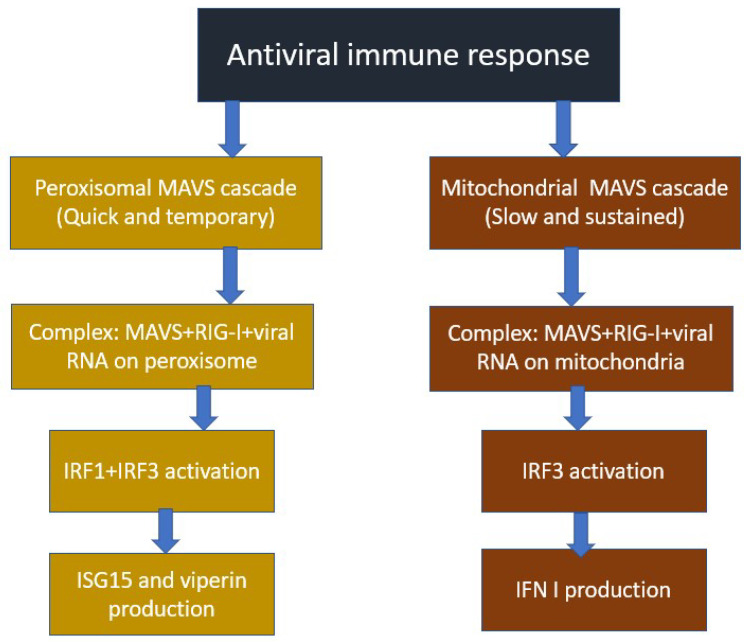
Mitochondrial and peroxisomal MAVS systems.

**Figure 2 cells-12-01072-f002:**
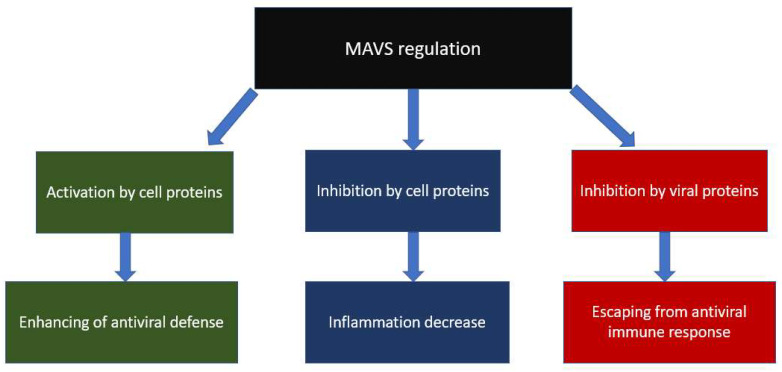
Different MAVS regulation.

**Figure 3 cells-12-01072-f003:**
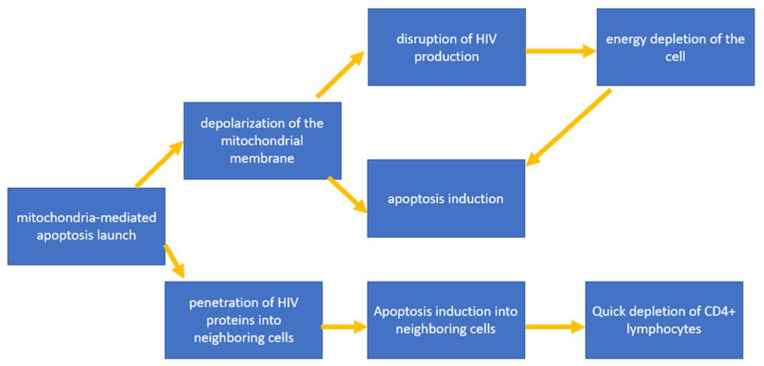
General scheme for triggering mitochondria-mediated apoptosis induced by HIV proteins.

**Figure 4 cells-12-01072-f004:**
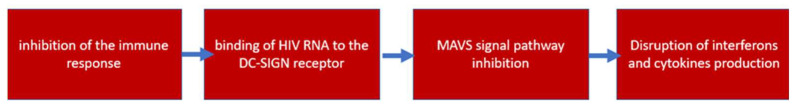
Scheme of inhibition of the immune response against HIV.

**Figure 5 cells-12-01072-f005:**
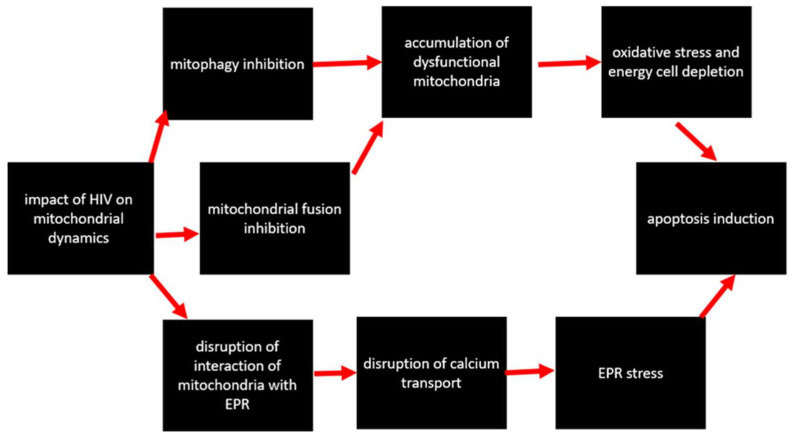
Impact of HIV on mitochondrial dynamics.

**Table 1 cells-12-01072-t001:** Impact of ART drugs on the development of mitochondrial dysfunction in HIV.

Drug Group	Mechanisms of Action on HIV	Mechanisms of Development of Mitochondrial Dysfunction
NRTI	Disruption of the HIV reverse transcription process with integration into HIV DNA and initiation of chain break	Mutagenesis and inhibition of mitochondrial DNA synthesis; Disruption of ATP/ADP translocation; Decreased expression of cytochrome c oxidase; Development of oxidative stress; Disruption of oxidative phosphorylation.
NNRTI	Disruption of the HIV reverse transcription process by direct inhibition of reverse transcriptase	Decreased mitochondrial membrane potential followed by rupture; Disruption of oxidative phosphorylation; The release of cytochrome C into the cytoplasm with the initiation of apoptosis.
IP	Inhibition of maturation of HIV proteins by direct action on HIV integrase	Decreased mitochondrial membrane potential followed by rupture; Disruption of oxidative phosphorylation; The development of oxidative stress, both as a result of their production by mitochondria, and by an increase in the expression of NADH oxidase; Disruption of glucose transport and lipid metabolism; The release of cytochrome C into the cytoplasm with the initiation of apoptosis.

## Data Availability

Not applicable.

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
