# Peer review of "Impaired Mitochondrial Function in T-Lymphocytes as a Result of Exposure to HIV and ART"

_cells, 2023, doi:10.3390/cells12071072_

Round 1

Reviewer 1 Report

There are few english errors, examples Line 81 "immigrate" should be "migrate". Line 89 "a hypothesis" should be "an hypothesis". Line 96 "defeat" should be "reduction", Line 97 "Violation" should be "change" or similar word.  Line 267 "to to" to doubled up. Section 3.2/3 This needs to include discussion of peroxisomal MVAS as well as the mitochondrial MVAS. See DOI: 10.1016/j.cell.2010.04.043 (Viral defense: It takes two MVAS to tango) plus other literature. A diagram showing the components of the anti-viral MVAS pathways would be worthwhile considering. Section 4.3 is a little speculative with less evidence than the previous sections of the paper. Section 6 Discussion. This should be retitled as it is not a classical discussion related to findings in the paper. I would use something like "future research" or something similar.

Author Response

Response to Reviewer 1 Comments

Point 1:  There are few english errors, examples Line 81 "immigrate" should be "migrate". Line 89 "a hypothesis" should be "an hypothesis". Line 96 "defeat" should be "reduction", Line 97 "Violation" should be "change" or similar word.  Line 267 "to to" to doubled up.

Response 1: Correction was made except "a hypothesis" should be "an hypothesis" because right article was used.

Point 2: Section 3.2/3 This needs to include discussion of peroxisomal MVAS as well as the mitochondrial MVAS. See DOI: 10.1016/j.cell.2010.04.043 (Viral defense: It takes two MVAS to tango) plus other literature. A diagram showing the components of the anti-viral MVAS pathways would be worthwhile considering.

Response 2: The text and new figure were added in the end of 3.2. Section.

Point 3:  Section 4.3 is a little speculative with less evidence than the previous sections of the paper.

Response 3: Additional references to studies (42,43) on this topic have been added to this section.

Point 4: Section 6 Discussion. This should be retitled as it is not a classical discussion related to findings in the paper. I would use something like "future research" or something similar.

Response 4: It was retitled.

Reviewer 2 Report

The manuscript entitled “Impaired mitochondrial function in T-lymphocytes as a result of exposure to HIV and ART" considers the role of the development of  mitochondrial dysfunction in the progression of HIV infection and the onset of AIDS, as well as the direct impact of HIV on mitochondria. However, the presented article could not be published in Cells for the following.

1.      Introduction is so week. The purpose of the current review article should be justified. More details about the topic should be discussed and enriched with recent references.

2.      Discussion part seems to be very short and needs more work to be modified.

3.      Abbreviations should be spelled out in the title and abstract such as antiretroviral therapy (ART).

4.       More tables and figures should be added to enrich the article.

5.      Finally, English should be polished throughout the text

Author Response

Response to Reviewer 2 Comments

Point 1: Introduction is so week. The purpose of the current review article should be justified. More details about the topic should be discussed and enriched with recent references.

Response 1: Additional information was added in Introduction.

Point 2: Discussion part seems to be very short and needs more work to be modified.

Response 2: The Discussion section, which was renamed "Future research" at the request of another reviewer, has been expanded.

Point 3:  Abbreviations should be spelled out in the title and abstract such as antiretroviral therapy (ART).

Response 3: Abbreviations were spelled out in the abstract.

Point 4: More tables and figures should be added to enrich the article.

Response 4: The new figure 1 and 2 were added.

Point 5: Finally, English should be polished throughout the text

Response 5: The found mistakes were corrected.

Round 2

Reviewer 2 Report

The manuscript could be published in its present form